# Predicting Ophthalmologist Gaze Patterns on OCT Data with Masked Autoencoders and Long Short-Term Memory Networks

Tri Le[1], Kuang Sung[2], Kaveri A. Thakoor[123]

[1]Department of Computer Science, Columbia University, New York, NY, United States

[2]Department of Biomedical Engineering, Columbia University, New York, NY, United States

[3]Department of Ophthalmology, Columbia University Irving Medical Center, New York, NY, United States
{tql2000, ks3864, k.thakoor}@columbia.edu

*Abstract*—Scanpath prediction is crucial in the medical domain as it captures the visual attention patterns of experienced clinicians, offering insights into diagnostic processes and enhancing training programs. Understanding where experts focus can lead to improved medical imaging interpretation and decision-making. However, scanpath prediction is extremely challenging due to the inherent noise in eye-tracking data, individual variability among clinicians, and the complexity of medical images.

This work introduces a pioneering adaptation of the "Show, Attend and Tell" (SAT) [1] framework to analyze the gaze patterns of ophthalmologists on Optical Coherence Tomography (OCT) reports. Instead of using Convolutional Neural Networks (CNNs) for visual feature extraction, we integrate self-supervised learning through a Masked Autoencoder (MAE) [2]. The MAE reconstructs masked regions of OCT images, enabling the encoder to generate robust image representations despite limited labeling in medical imaging datasets. We trained separate LSTM models for each clinician to account for individual inspection patterns.

The model demonstrated strong evaluation results, with the best-performing model achieving a ScanMatch score up to 0.5595 and Pearson correlation of up to 0.866 in predicting expert gaze on OCT reports. We showcase a downstream use-case of predicting the sequence of expert-fixated regions on an OCT report and visualizing these for ophthalmic resident education. Our findings highlight the framework's potential to enhance the understanding and emulation of expert-level diagnostic mechanisms, aiding in the explanation of AI-based predictions in the clinic and guiding novice residents in ophthalmic education, especially in resource-diverse environments with limited access to expert ophthalmologists or labeled datasets.

*Index Terms*—Attention, Deep Learning, Scanpath Prediction, Gaze Prediction, LSTM, Ophthalmology, Optical Coherence Tomography, Transformer, Vision Transformer

## I. INTRODUCTION

IN the realm of medical imaging, particularly Optical Coherence Tomography (OCT), the interpretation of complex visual data plays a crucial role in diagnostic processes. Eye tracking data has been a great source of information in medical applications of machine learning including detection [10], [12] and segmentation [11]. Previous works have successfully integrated clinician's gaze information to inform Vision Transformer [3] based models for both OCTs [7] and X-ray images [8], enhancing both detection accuracy and computational efficiency. Expert sonographers' gaze during ultrasound image inspection is used for model compression without significant loss in performance [9]. These examples show that eye gaze from seasoned clinicians carries a wealth of information across different types of medical imaging. However, limited work has attempted to predict experienced ophthalmologists' gaze fixation location and order during image inspection for downstream diagnostic and educational purposes. Scanpath prediction, which forecasts the sequence of eye movements during image viewing, has been extensively studied in natural images [16]–[27]. However, its application to medical imaging remains underexplored. To the best of our knowledge, we are the first to develop a method for scanpath prediction in Optical Coherence Tomography (OCT) reports that is also agnostic to different types of medical images.

Through years of experience, expert ophthalmologists develop nuanced gaze patterns that guide their image analysis and diagnoses. However, the scarcity of experienced ophthalmologists poses a significant challenge, especially in remote or under-served areas. This project seeks to bridge this gap by leveraging the "Show, Attend, and Tell" [1] framework, traditionally used for image captioning, to predict gaze patterns on OCT reports using gaze data from experienced ophthalmologists. Such a predictive model could serve as a virtual advisor to novice ophthalmologists, guiding their inspection of OCT reports in the absence of attendings or experienced doctors, or as a second opinion/corroboration even for expert doctors in ambiguous cases.

A novel aspect of this project is the replacement of the conventional Convolutional Neural Network (CNN) encoder with an encoder derived from a Masked Autoencoder, specifically trained on OCT images. This approach exploits the advantages of self-supervised learning, which does not rely on labeled data. This is particularly beneficial in the medical

imaging domain, where acquiring labeled data is not only challenging but also labor-intensive. Through this innovation, we aim to harness extensive, unlabeled OCT datasets, thereby overcoming one of the significant hurdles in medical image analysis. The integration of self-supervised learning into our predictive model promises to enhance the accessibility and accuracy of OCT report interpretations, democratizing expert-level diagnostic guidance across the medical community.

## II. Related Work

### A. Masked Autoencoders

In deep learning, the Masked Autoencoder (MAE) represents a significant step up in the context of self-supervised learning and representation learning in the computer vision domain. Originally conceptualized to address the inefficiencies and data constraints of supervised learning, MAE has rapidly gained traction across various domains, including medical imaging [4]. At its core, the MAE solves the pretext task in the image domain by intentionally obscuring parts of the input data, compelling the model to predict these missing segments based solely on the available information. This methodology not only enhances the model's feature extraction capabilities but also its generalization to unseen data, making it particularly suited for domains where labeled data is scarce or expensive to obtain.

In medical imaging, specifically in the analysis of Optical Coherence Tomography (OCT) images, the application of MAE has opened new vistas. The encoder component of MAE, adept at capturing intricate details from partial data, has shown promise in understanding complex visual patterns intrinsic to medical diagnostics. Studies leveraging MAE for natural images highlight its superiority in extracting relevant features without explicit labeling, showcasing its potential to be applied to medical image interpretation, specifically the prediction of expert gaze on OCT images.

### B. Neural Image Captioning with Visual Attention

The "Show, Attend, and Tell" framework [1], initially introduced for image captioning tasks, has established itself as a cornerstone in computer vision and natural language processing. Its innovative use of attention enables models to focus on specific parts of an image while generating corresponding textual descriptions, mimicking the human ability to correlate visual elements with linguistic annotations. This approach has not only enhanced the interpretability of deep learning models but also has significantly improved their performance by allowing them to prioritize salient features in images.

Adapting the "Show, Attend, and Tell" methodology to the domain of scanpath prediction, particularly within medical imaging, marks an effort to translate visual attention mechanisms from textual annotation to the prediction of visual focus areas. In the context of analyzing Optical Coherence Tomography (OCT) reports, this adaptation could change how clinicians engage with medical imagery. By predicting where an experienced ophthalmologist's gaze might linger on an OCT image, this framework offers a tool for training and assisting less experienced clinicians, while also potentially providing corroboration/confirmation for the reasoning behind a diagnosis for experts. This application of "Show, Attend, and Tell" in scanpath prediction underscores the versatility of attention mechanisms and highlights the potential of machine learning to enhance diagnostic practice within ophthalmology.

## III. Methodology

The original "Show, Attend, and Tell" framework utilizes an LSTM to generate a probability distribution over a vocabulary through softmax, framing the problem as one of classification. In this study, we compared two independent approaches to capture the expert gaze patterns that trained the LSTM to be an autoregressive model and a classification model. We used the autoregressive version as a convenient sanity check to make sure that the model could capture the gaze pattern and that the attention network (fully connected layers) used during the training of the LSTM is paying attention to the critical regions of OCT reports. We then used qualitative results of the autoregressive model as a premise to re-frame the training as a classification problem which is the main application of this study.

This study also examines the robustness of the SAT framework through the substitution of its conventional CNN encoder with the encoder from a Masked Autoencoder (MAE). While CNNs have been the standard for extracting image representations, they are inherently suited for classification tasks that require labeled data. In the SAT framework, obtaining image embeddings is an intermediary step that does not necessarily rely on labels, making the MAE, which can learn from unlabeled data, an apt choice for medical imaging contexts where labels may be scarce. This shift not only challenges the traditional reliance on CNNs but also aligns with the evolving needs of medical image analysis to learn from extensive, unlabeled datasets.

More about model training is explained in the sub-section III-B.

### A. Datasets

In this study, two types of internal datasets were utilized, both centred around Optical Coherence Tomography (OCT) reports. The first dataset is comprised of approximately 13,000 OCT reports (6,000 labeled as glaucomatous vs. not glaucomatous and 7000 labeled as acceptable vs. unacceptable, with some overlap with the 6000-dataset and including poor scans), which were used to conduct self-supervised learning on the Masked Autoencoder (MAE). The second dataset includes gaze patterns from seven experienced ophthalmologists, collected while each expert examined 20 OCT reports. Each fixation in each pattern is in normalized coordinates with respect to computer screen resolution. The 20 OCT reports (per clinician) were randomly sub-selected from a larger pool of OCT reports collected between 2010 and 2023 at our institution (through an IRB-approved study: Protocol AAAU4079 approved by the Columbia University Irving Medical Center Institutional Review Board). A different set of 20 reports was

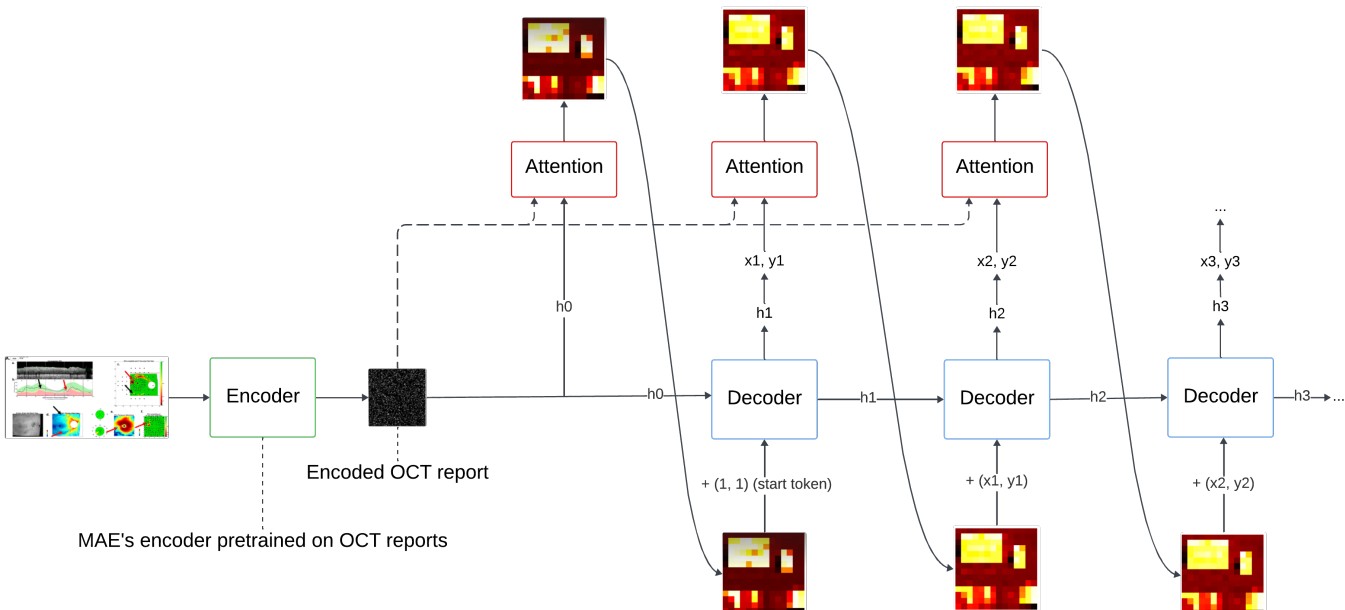

Fig. 1: Overall pipeline for capturing gaze pattern. The OCT first goes through the MAE's encoder to output the encoded image. This encoding is used along with LSTM's hidden state for each time step to calculate the attention heatmap. Gaze information includes x, y as coordinates.

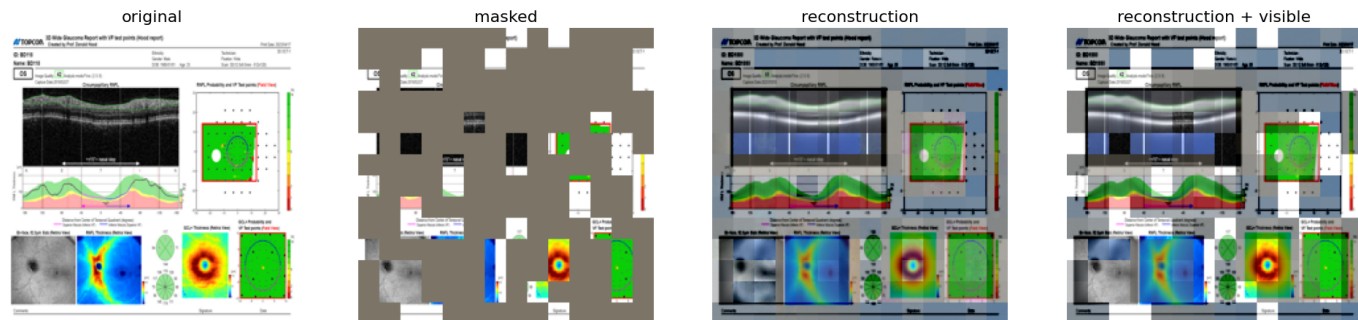

Fig. 2: MAE's reconstruction results on OCT report. In order from left to right, the first image is the original OCT report. The second image is the randomly masked version of the OCT report that is used as input during MAE's training. The third image is the reconstruction result of both visible and masked patches. The fourth image is the combination between the visible masks and reconstructed patches that were masked.

used for each clinician; however, due to small dataset size, there was some overlap in reports viewed by clinicians; on average, each report was viewed by 3 clinicians. For each LSTM model corresponding to a clinician, we trained on 15 gaze patterns and tested on 5 gaze patterns. We also performed 4-fold cross-validation on each LSTM to retrieve the 95% confidence interval to ensure statistical significance.

### B. Model

Fig. 1 illustrates the real-time fixation prediction pipeline. In the inference stage, an OCT report is initially processed by the MAE encoder to extract the image representation. This representation, along with the LSTM's hidden state, is then fed into the attention network, generating an attention heatmap. Utilizing this heatmap and gaze information at each timestep, the model predicts the coordinates of the next gaze. The study encompasses two primary training phases, outlined below.

*1) Training Mask Autoencoder:* The MAE is trained on 13,000 OCT reports to learn data representation. Initially, it randomly masks 75% of the patches from an OCT report. The remaining visible patches are processed through 12 Vision Transformer (ViT) [3] blocks during the encoding phase. The encoder's output, or image embedding, combined with mask tokens (place holders for masked patches at the beginning), is input into a decoder comprising 8 ViT blocks. The decoder's output reconstructs the masked OCT report, as depicted in Fig. 2. A more powerful or deeper decoder is unnecessary for the MAE, as we only employ its encoder for our primary application.

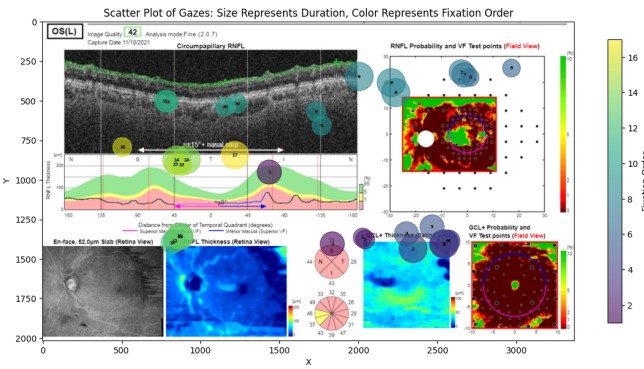

(a) Original fixations of the clinician on report A

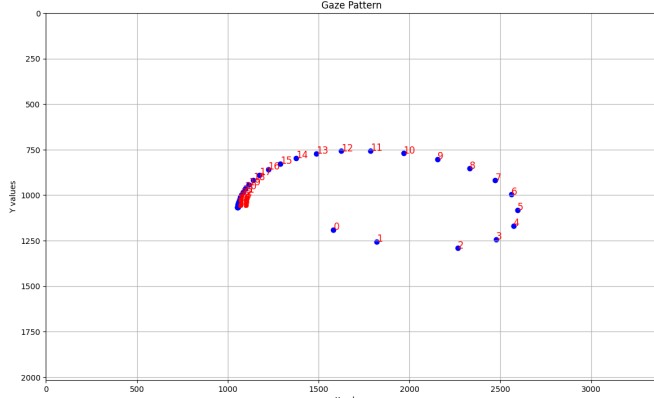

(b) Gaze pattern predicted by the model for report A

Fig. 3: A report with ground truth fixations (left column 3a) by the same clinician and their corresponding predicted gaze patterns by the auto-regressive version of the model (right column 3b). One can observe a pattern where the inspection begins at the report's center and proceeds in an anti-clockwise direction. The autoregressive model could reproduce the low-variance version of such patterns, as shown on the right.

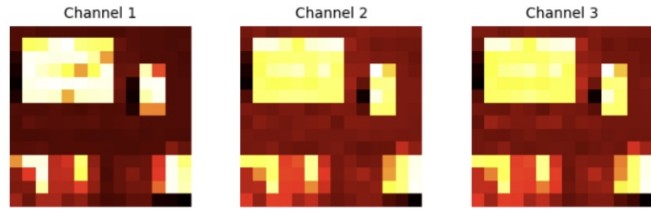

Fig. 4: First three attention heatmaps used by the model to predict gaze coordinates at each time step (worded by channel number) for report 3a. The brighter (yellow) the region, the more attention is paid by the model to predict the gaze coordinates.

*2) Training LSTM:* To train the LSTM within the SAT framework, encoding the image is essential, a task undertaken by the MAE as previously mentioned. During each training timestep, the LSTM's hidden state is merged with the encoded image and processed by an attention network to produce an attention heatmap. This heatmap guides the LSTM to focus on critical regions for predicting the next gaze's coordinates. The heatmap, alongside the previous gaze prediction, is used to forecast the subsequent gaze information. To avoid error propagation to later steps and ensure each prediction relies on accurate historical data, teacher forcing is employed during the training phase which is akin to language model training. As we have a limited number of scanpaths per clinician, we employ early stopping, dropout and gradient clipping as regularization strategies to prevent overfitting. As proposed in the original work of Show, Attend, Tell [1], we inherently use doubly stochastic regularization to make sure the model pays attention equally to all critical sub-regions of the OCT report during the generation the next fixation coordinates.

*a) Training the LSTM as an autoregressive model:* Framing training the LSTM as a regression problem requires gaze information (2D normalized coordinates) for both input and output at each LSTM time step to be continuous. The normalized coordinates are concatenated with the heatmap to become the complete input at each time step. During inference, output at the current time step is used as the input for the next time step.

*b) Training the LSTM as a classification problem:* When training the LSTM as a classification problem, we divided the OCT report into a 7x7 grid. One-hot encoding is used to represent the input gaze coordinates. The one-hot encoding representation of gaze is formed by first multiplying the normalized coordinates by the grid's size to arrive at which grid cell the gaze falls into. The coordinate of a grid cell is then flattened to form the one-hot representation. Output for each LSTM step is the logits for 49 cells, and softmax is applied to retrieve the probability distribution among all cells. During inference, unlike the autoregressive LSTM where error propagation is inherent, we used beam search to significantly remedy this issue which yields a more globally optimal gaze pattern.

*3) Beam Search:* Beam search is a heuristic search algorithm used in natural language processing for decoding sequences. It explores multiple possible outputs by maintaining a fixed number of best partial hypotheses (beams) at each step, enhancing the likelihood of generating the most probable sequence in tasks like machine translation.

In gaze pattern prediction using LSTM, beam search mitigates error propagation by evaluating multiple gaze paths simultaneously. This approach retains the most promising trajectories, preventing early mistakes from significantly influencing future predictions, thereby enhancing the accuracy and reliability of predicting gaze sequences over time.

*4) Consideration and Simplification:* During the training of the MAE, OCT reports were resized to a square aspect ratio. Gaze coordinates were not affected by this change since

they were originally normalized based on the image resolution during the data collection phase. During training of the LSTM on gaze data, we only used the MAE's encoder to extract OCT report embeddings. Since the MAE's decoder is used to reconstruct masked images, it cannot be utilized in this application.

## C. Evaluation

We conducted both quantitative and qualitative evaluations of the model's predictions to ensure its ability to learn and replicate the gaze patterns of clinicians. To measure the spatial alignment and consistency between the model's output fixations with the regions deemed critical by clinicians during their inspections, we used the Pearson Correlation on saliency maps [14]. To assess the similarity between the predicted and actual scanpaths, various methods are available [15]. We opted for ScanMatch [13], which effectively captures both the positional and temporal aspects of scanpaths [15].

*1) Saliency maps consistency with Pearson Correlation:* Previous work [5] used Pearson correlation coefficients to quantitatively evaluate the consistency of the regions of interest (ROIs) identified during fundus image inspections for glaucoma diagnosis by different ophthalmologists. They computed these coefficients between Gaussian-filtered fixation maps, with each map representing the gaze pattern of an individual ophthalmologist. We took a similar approach with a slight modification to evaluate how well the model performs in learning to capture individual clinician gaze patterns and their focused regions during the inspection. Detailed steps are explained below.

The smoothed grid $S$ is retrieved by applying the Gaussian filter with a standard deviation set to one on the original binary grid cell $I$. This is done on both predicted and ground-truth gaze patterns.

$$S = I * G \tag{1}$$

After smoothing, the smoothed grid is converted into a vector format:

$$\mathbf{s} = [S_{11}, S_{12}, \ldots, S_{17}, S_{21}, \ldots, S_{77}] \tag{2}$$

The Pearson correlation coefficient between the flattened ground truth and prediction grids is calculated as:

$$r = \frac{\sum_{i=1}^{n}(s_{gt,i} - \bar{s_{gt}})(\hat{s}_i - \bar{\hat{s}})}{\sqrt{\sum_{i=1}^{n}(s_{gt,i} - \bar{s_{gt}})^2 \sum_{i=1}^{n}(\hat{s}_i - \bar{\hat{s}})^2}} \tag{3}$$

where $n = 49$ (since the grid is 7x7), $\mathbf{s_{gt}}$ is the flattened ground truth grid, and $\hat{s}$ is the flattened prediction grid.

*2) Scanpaths similarity with ScanMatch:* ScanMatch is a method designed to assess the similarity between scanpaths, utilizing the Needleman-Wunsch algorithm, originally developed for DNA sequence comparison in bioinformatics [13]. This approach involves spatially and temporally binning saccadic eye movement sequences, which are then recoded into sequences of letters that encapsulate information about

fixation locations and order. The method evaluates the similarity between these sequences by maximizing a similarity score derived from a substitution matrix. This matrix scores all possible letter pair substitutions and incorporates a penalty for gaps. Scanpath prediction models widely adopt ScanMatch for evaluation due to its robustness to inherent noise in saccadic eye movements [18].

## D. Integration into Unity for GUI for Clinical Translation

Given OCT-report patches predicted by our algorithm, we developed a user interface (UI) using Unity, a versatile cross-platform game engine, to present a sequence of bounding boxes on OCT reports. The model's prediction, a sequence of coordinates on a 7x7 grid, is translated into corresponding bounding boxes and overlaid on top of the OCT reports. The UI facilitates the visualization of annotated regions of interest, allowing users to interactively explore and analyze spatial and temporal relationships between regions of interest (ROIs) within OCT data.

## IV. RESULTS & DISCUSSION

| Clinician number | With Beam Search | Without Beam Search |
|---|---|---|
| 1 | 0.8659 ± 0.0735 | 0.4065 ± 0.2040 |
| 2 | 0.8070 ± 0.1180 | 0.5713 ± 0.3090 |
| 3 | 0.7164 ± 0.0654 | 0.4605 ± 0.1420 |
| 4 | 0.7041 ± 0.1250 | 0.5539 ± 0.1680 |
| 5 | 0.6626 ± 0.0939 | 0.4728 ± 0.1520 |
| 6 | 0.8072 ± 0.0876 | 0.6870 ± 0.1270 |
| 7 | 0.7626 ± 0.1310 | 0.5247 ± 0.1570 |

TABLE I: Pearson correlations with 95% confidence intervals, evaluated by 4-fold cross-validation across all clinicians engaged in the study. Scanpath prediction post-processed by beam search quantitatively shows superior results.

| Clinician number | With Beam Search | Without Beam Search |
|---|---|---|
| 1 | 0.3269 ± 0.1027 | 0.2652 ± 0.0367 |
| 2 | 0.3522 ± 0.0658 | 0.2405 ± 0.0369 |
| 3 | 0.3352 ± 0.0505 | 0.2665 ± 0.0395 |
| 4 | 0.3269 ± 0.0953 | 0.2668 ± 0.0514 |
| 5 | 0.3690 ± 0.0729 | 0.2513 ± 0.0318 |
| 6 | 0.5224 ± 0.0370 | 0.3172 ± 0.0278 |
| 7 | 0.5595 ± 0.0640 | 0.3326 ± 0.0306 |
| **Mean** | 0.4032 | 0.2772 |

TABLE II: The ScanMatch metric with 95% confidence intervals and evaluated through a 4-fold cross-validation among all clinicians involved in the study. The gaze pattern, refined using beam search, shows quantitatively better outcomes in resembling the clinician's gaze patterns.

## A. Mask Autoencoder on OCT report

Fig. 2 shows the reconstruction result of MAE trained on OCT reports. The training of the Masked Autoencoder (MAE) on 13,000 Optical Coherence Tomography (OCT) reports yielded qualitatively impressive results demonstrating its transferability from natural images to medical image applications.

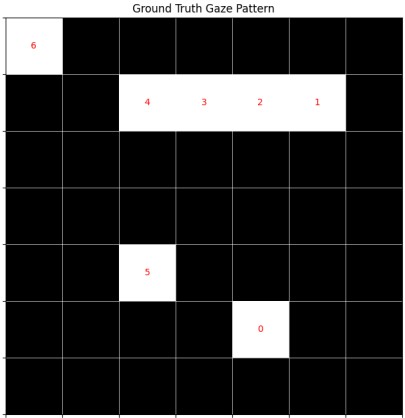
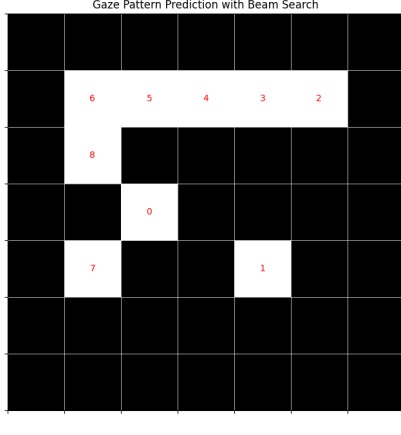
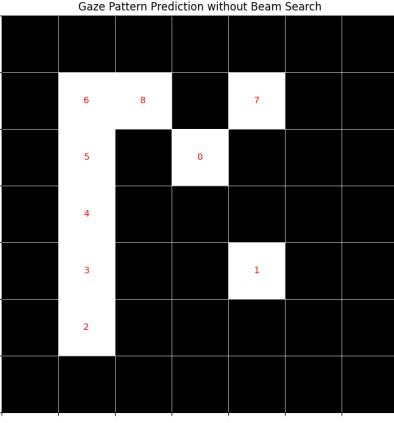

| (a) Ground Truth Gaze Pattern | (b) Predicted Pattern with Beam Search | (c) Predicted Pattern without Beam Search |
|---|---|---|

Fig. 5: Qualitative results for LSTM trained as a classification problem on 7x7 grid cells. Testing on the same report, beam search can significantly reduce error propagation to output the more globally optimal gaze pattern resembling the ground truth gaze pattern of the experienced ophthalmologist.

This further shows its potential as a viable encoder within the "Show, Attend, and Tell" framework. By effectively learning the data representation, the MAE showcases its capability to capture the intricate details and nuances present in OCT images. This proficiency not only enhances the model's understanding of the visual content but also significantly improves the subsequent attention-driven processes.

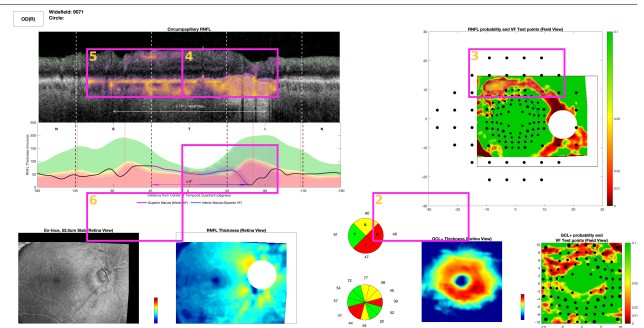

Fig. 6: Unity interface displaying an OCT report with overlaid bounding boxes and a heatmap. ViT-generated regions of interest are highlighted in orange, and our LSTM-predicted ordering of ROIs are shown by numbers in left corner of each box.

### B. LSTM trained as an autoregressive model

Fig. 3 displays the model's gaze pattern predictions for two distinct reports. On closer inspection of the ophthalmologist's gaze sequence (3a), one can observe a pattern where the inspection begins at the report's center and proceeds in an anti-clockwise direction. The model successfully captures these patterns, as depicted to the right (3b). Intriguingly, the model appears to have learned a regression function tailored to each report's gaze points, resulting in a series of discrete points that follow a smooth, complex curve. While these results affirm

the method's effectiveness and provide insights into an expert clinician's gaze pattern, they exhibit lower variance compared to the actual collected gaze data. The model's predictions do not isolate specific regions within the OCT report that could potentially assist less experienced ophthalmologists.

In Fig. 4, the attention mechanism's role in predicting gaze patterns for report 3a and its associated gaze pattern 3b is evident. Notably, the LSTM's attention network identifies key areas (indicated by brighter regions) in the OCT report, such as the circumpapillary retinal nerve fiber layer (RNFL) scan—visible as a grayscale rectangular image in the report's upper left. This validates the effectiveness of the MAE in supplanting traditional CNNs within the conventional SAT framework. Whereas, in the original SAT framework for image captioning, the attention is sharply focused—for instance, on a balloon when predicting the word "balloon"—the adapted framework for gaze prediction utilizes all relevant areas on the OCT image to anticipate the next gaze point.

### C. LSTM trained as a classification model

*1) Saliency maps consistency with Pearson Correlation:* Table I shows promising results on seven different seasoned ophthalmologists. The Pearson correlation calculated on the hold-out test set shows a strong correlation between prediction and ground-truth gaze patterns. Moreover, beam search serves a critical role which helps retrieve the more globally optimal gaze pattern. This could be qualitatively visualized in Fig. 5.

Compared to the autoregressive version of the model, more variance in the pattern is introduced in the LSTM trained as a classification problem. The predicted pattern post-processed by beam search showed a clear pattern that resembles the ground truth gaze pattern of the experienced clinician.

*2) Scanpaths similarity with ScanMatch:* Table II shows the results for ScanMatch metric for the same seven clinicians. Beam search, again, further improves this metric by reducing

| Method | AiR (VQA) | OSIE (free viewing) | COCO-Search18 (visual search) | MASSVIS |
|---|---|---|---|---|
| PathGAN [16] | 0.210 | 0.077 | 0.277 | 0.232 |
| SaltiNet [17] | 0.112 | 0.169 | 0.199 | 0.331 |
| Chen et al. [18] | **0.394** | **0.383** | **0.554** | – |
| IOR-ROI [19] | 0.171 | 0.267 | 0.316 | – |
| Itti et al. [20] | – | 0.211 | – | – |
| SGC [21] | – | 0.211 | – | – |
| Wang et al. [22] | – | 0.151 | – | – |
| Le Meur et al. [23] | – | 0.228 | – | – |
| STAR-FC [24] | – | 0.204 | – | – |
| IRL [25] | – | – | 0.403 | – |
| DCSM [26] | – | – | – | 0.328 |
| UMSS [27] | – | – | – | **0.387** |

TABLE III: Mean ScanMatch values of different methods across different tasks and datasets. The best performant model for each dataset is bolded. Compiled results retrieved mainly from [18], [27].

error propagation during LSTM inference and producing scanpath patterns that more closely resemble those of clinicians.

As we are the first to develop a model for scanpath prediction on medical images, particularly for ophthalmic OCT reports, there are no standardized metrics to perform direct benchmarking on this data. However, the ScanMatch score is used for benchmarking scanpath prediction on natural images and accounts for both spatial and temporal aspects of gaze. We therefore compiled Table III with results from previous works on scanpath predictions on different tasks and datasets of natural images. Although the comparison does not fully establish the superiority of our model over others due to significant differences between natural and medical images, it demonstrates our model's strong performance and provides a baseline for future research.

AiR [28] is a Visual Question Answering dataset consisting of images and questions, as well as eye-tracking data from 20 participants. OSIE [29] is a free-viewing dataset including 700 images with eye-tracking data from 15 participants. COCO-Search18 [25] is a visual search dataset including images annotated with the fixations of people searching for target-object goals. MASSVIS [30] is a popular dataset covering various types of visualizations which also provides gaze data recorded from human viewers.

Our model's mean ScanMatch metric from Table II shows a competitive result (0.4032) compared to other models trained on natural images in Table III. While Chen et al. [18] reaches a ScanMatch score of 0.554 in the COCO-Search18 dataset, we argue that scanpaths of experienced clinicians on OCT reports inherently differ from those of viewers searching for target objects in natural images. Natural objects are usually concrete and well-segmented. This is not the case in medical images (e.g., OCT reports) which may contain ambiguous sub-regions that are critical for making diagnoses.

### D. Integration into Unity for GUI for Clinical Translation

As shown in Fig 6, we used Unity to develop an interactive UI that overlays a sequence of bounding boxes on OCT reports. Our model provides temporal information in the form of a sequence of bounding boxes based on a 7 by 7 grid. This

temporal information is combined with a heatmap generated by a Vision Transformer (ViT) through attention rollout [7]. By intersecting the heatmap with the temporal bounding boxes, we merge the temporal insights from our model with the attention mechanism of the ViT, enhancing the precision and informativeness of the annotated regions. Our results indicate that this approach effectively highlights key areas within the OCT reports, facilitating detailed analysis. Our preliminary user studies also show that the guidance can enhance efficiency of novices during inspection of OCT reports by reducing the variability of their scanpaths without compromising their diagnostic accuracy. This development underscores the potential of combining temporal and attention mechanisms to enhance the interpretability of complex medical data, providing a robust tool for clinical diagnostics and education.

### E. Limitations

We observe that some of the predicted bounding boxes covered less informative white regions. We suspect that this behavior partially stems from fixations located in white space that are part of clinician saccades during data collection. In future work, we plan to perform segmentation on OCT images and use segmentation masks to eliminate fixation points that fall into white space.

When clinicians repeatedly check the same region of OCT scans, it indicates the critical nature of the sub-regions. Our current LSTM model struggles to capture this recurrent inspection behavior effectively. To address this in future work, we will synthesize a heatmap for each OCT report of each clinician to indicate the importance of sub-regions based on the frequency of inspections. This heatmap would then serve as a supervised signal, allowing the LSTM to predict fixation coordinates and assign an importance score to each sub-region at every timestep. This enhancement would be especially useful for training novices, directing their attention to areas of significance and enhancing the model's interpretability.

### V. CONCLUSIONS AND FUTURE DIRECTIONS

This project presents a novel translation of the "Show, Attend, and Tell" (SAT) framework for predicting ophthalmologist gaze patterns on OCT reports, through the innovative

use of a Masked Autoencoder (MAE) as the encoder. This adaptation leverages self-supervised learning, allowing for a more nuanced interpretation of complex OCT images without the need for extensive labeled data.

Initial testing within the SAT framework, utilizing regression to anticipate gaze patterns, demonstrates the model's proficiency in identifying the sequence of an ophthalmologist's examination. The attention network trained along with the LSTM is also able to identify critical regions in the OCT report. However, the practical application of the autoregressive LSTM is constrained byscanpath predictions which do not adequately reflect the actual variability in human observation. Higher variance in gaze patterns is introduced by training the LSTM as a classification problem. Post-processing of predicted gaze patterns using beam search yields patterns that better resemble the ground truth and achieve quantitatively superior Pearson coefficients and ScanMatch metrics. Our model shows strong correlation between predicted and ground-truth gaze patterns, offering potential for conducting user studies with our Unity-based guidance interface to determine its effectiveness in guiding ophthalmologists during OCT examination. Future directions include integrating the temporal aspect of our work into a Vision Language Model (with clinician speech/dictations) to refer and ground features in medical images.

## ACKNOWLEDGMENTS

The authors are grateful to Richard Zemel for his insightful feedback and inspiration from his past work, to Mary Durbin from Topcon Healthcare for data sharing, to Donald C. Hood and Emmanouil Tsamis for their expert OCT report grading, and to Jeffrey M. Liebmann and George A. Cioffi for their oversight and guidance.

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
