# OpenReview forum: "Predicting Ophthalmologist Gaze Patterns on OCT Data with Masked Autoencoders and Long Short-Term Memory Networks"
_IEEE.org/EMBS/BHI/2024/Conference — IEEE BHI'24_

### Official Review · Reviewer_9WUk · 2024-08-10
**Review of "Predicting Ophthalmologist Gaze Patterns on OCT Data with Masked Autoencoders and Long Short-Term Memory Networks"**

**Overall Rating:** 6
**Confidence:** 4

**Other Quality Metrics:**

Clarity of writing: Good
Clinical Significance: Great
Methodological Novelty: Great
Experiments and Results: Good

**Questions For The Authors:**

What are some advantages of choosing a LSTM-based model over a transformer-based one? Since the encoder already integrates self-supervised learning, why not include it in the decoding as well? Is the choice motivated by data availability, compute, or the clinical demand for interpretability?

**Strengths:**

1. The clinical motivation to adapt the SAT framework is novel, and the use of OCT data is innovative.
2. The model design for masked autoencoder + LSTM decoder and presented with a clear clinical rationale. The experiments that compare LSTM as a regression vs classification model is detailed and thorough.
3. The paper presents a clinically grounded metric that builds on the Pearson correlation coefficient to evaluate the consistency of the predicted areas of interest.

**Summary Of The Paper:**

This paper presents a novel AI-based approach to predict the gaze patterns of experienced ophthalmologists on optical coherence tomography (OCT) reports. Specifically, it adapts the "show, attend, tell" framework by leveraging a masked auto-encoder for reconstructing regions of interest, and a LSTM model to perform gaze prediction based on the patterns of each clinician. In addition, the paper borrows the use of beam search decoding from natural language processing to filter erroneous gaze pathways and improve the accuracy of the LSTM model in inference.

**Weaknesses:**

1. Lack of baselines. The paper mainly focuses on comparing different configurations of its own LSTM model, while it does not compare it with more recent transformer-based autoregressive models. The results would be more compelling if the performance of LSTM is comparable to that of state-of-the-art models.
2. Clarity of figures. Figure 4, for instance, should be zoomed in to better show the differences between each heat map, or show less heat maps. The current display makes them look the same.

---

### Official Review · Reviewer_BjdS · 2024-08-15
**Predicting Ophthalmologist Gaze Patterns on OCT Data with Masked Autoencoders and Long Short-Term Memory Networks**

**Overall Rating:** 8
**Confidence:** 5

**Other Quality Metrics:**

(a) Clarity of writing;
great

(b) Clinical Significance;
good

(c) Methodological Novelty;
good

(d) Experiments and Results
great

**Questions For The Authors:**

The authors claim excellent results. However, the correlations can be low. Can you please expand on this point?

Also, do you think that Pearson correlations capture the need for accuracy in this problem?

**Strengths:**

- The paper is relatively well-written and well-motivated.
- The prediction of gaze patterns is an important problem and the paper reports good progress on the problem.

**Summary Of The Paper:**

The paper presents a methodology for predicting opthalmologist gaze patterns on OCT data. The method is based on the use of masked autoencoders and LSTM networks. Overall, it looks like the approach worked well.

**Weaknesses:**

- The reconstructed images do not look very good. However, this does not matter as long as the overall results on the gaze patterns are good.

---

### Official Review · Reviewer_6woQ · 2024-08-17
**Good ideas but need to describe clinical relevance with experiments**

**Overall Rating:** 6
**Confidence:** 3

**Other Quality Metrics:**

(a) Clarity of writing - good
(b) Clinical Significance - fair
(c) Methodological Novelty - good
(d) Experiments and Results - fair

**Questions For The Authors:**

1. Dataset details: How were the 20 OCT reports per clinician selected? Were they chosen randomly or based on specific criteria?
Were the same 20 reports used for all clinicians, or did each clinician examine different reports? How representative are these 20 reports of the typical variety of cases an ophthalmologist might encounter?

2. What was the rationale behind choosing a 7x7 grid for the classification problem? How might different grid sizes affect performance?
How was the LSTM architecture (number of layers, hidden units, etc.) determined?

3. Why was Pearson correlation chosen as the primary evaluation metric? Were other metrics considered? How does the model's performance compare to human-level agreement between different expert ophthalmologists?

4. How well does the model generalize to OCT reports from different machines or institutions?
Can this approach be extended to other types of medical imaging beyond OCT?

**Strengths:**

1. The paper adapts the "Show, Attend and Tell" framework, originally used for image captioning, to predict expert gaze patterns on medical images. This is an innovative use of existing technology for a new purpose in healthcare.

2. By replacing the traditional CNN encoder with a Masked Autoencoder (MAE), the authors leverage self-supervised learning. This is particularly advantageous in medical imaging where labeled data can be scarce.

3. The model achieves high Pearson correlation coefficients (up to 0.866) between predicted and ground truth gaze patterns, indicating strong predictive power.

4. The use of beam search significantly enhances the model's performance, demonstrating the authors' efforts to optimize their approach.

5. The authors use multiple training approaches i.e., autoregressive and classification-based training for the LSTM, providing insights into different methodologies.

6. The practical application of the approach by developing a Unity-based interface for visualizing predicted gaze patterns shows a clear path to clinical translation of this research.

**Summary Of The Paper:**

The paper introduces a novel approach for predicting ophthalmologist gaze patterns on Optical Coherence Tomography (OCT) reports. The authors adapt the "Show, Attend and Tell" framework, replacing the traditional CNN encoder with a Masked Autoencoder (MAE) trained on OCT images. This allows for better feature extraction from medical images with limited labeled data. The model uses an LSTM to predict gaze sequences, trained both as an autoregressive model and a classification problem on a 7x7 grid. Results show strong correlations between predicted and ground truth gaze patterns, with beam search improving performance. The authors also developed a Unity-based interface to visualize the predicted gaze patterns on OCT reports. This work has potential applications in training novice ophthalmologists and providing second opinions for experts in ambiguous cases.

**Weaknesses:**

1. The study uses a relatively small dataset of only 20 gaze patterns per clinician, with 15 used for training and 5 for testing. This small sample size may limit the generalizability of the results. With such a small dataset and a complex model, there's a risk of overfitting. The paper doesn't discuss measures taken to prevent this, such as regularization techniques.

2. The paper doesn't compare their novel approach to existing methods for gaze prediction or medical image analysis, making it difficult to assess the relative improvement offered by this technique.

3. While the authors mention potential applications in training and second opinions, there's limited discussion on how the predicted gaze patterns translate to clinical outcomes or decision-making.

4. The autoregressive version of the model produces low-variance predictions that don't accurately reflect the variability in human gaze patterns. This limitation could reduce its practical usefulness. The paper doesn't provide a detailed analysis of where and why the model makes errors, which could provide insights for future improvements.

---

### Decision · Program_Chairs · 2024-09-23

Accept